# High Sensitivity Optical Fiber Mach–Zehnder Refractive Index Sensor Based on Waist-Enlarged Bitaper

**DOI:** 10.3390/mi13050689

**Published:** 2022-04-28

**Authors:** Na Zhao, Zelin Wang, Zhongkai Zhang, Qijing Lin, Kun Yao, Fuzheng Zhang, Yunjing Jiao, Libo Zhao, Bian Tian, Ping Yang, Zhuangde Jiang

**Affiliations:** 1State Key Laboratory for Manufacturing Systems Engineering, Xi’an Jiaotong University, Xi’an 710049, China; zn2020@xjtu.edu.cn (N.Z.); wzl15086927209@stu.xjtu.edu.cn (Z.W.); yao_kun@outlook.com (K.Y.); xjzfz123@stu.xjtu.edu.cn (F.Z.); 18830970939@163.com (Y.J.); libozhao@xjtu.edu.cn (L.Z.); t.b12@mail.xjtu.edu.cn (B.T.); ipe@xjtu.edu.cn (P.Y.); zdjiang@xjtu.edu.cn (Z.J.); 2Collaborative Innovation Center of High-End Manufacturing Equipment, Xi’an Jiaotong University, Xi’an 710054, China

**Keywords:** fiber optic sensor, Mach–Zehnder interferometer, optical fiber waist-enlarged bitaper, the refractive index sensitivity, corrosion

## Abstract

A Mach–Zehnder fiber optic sensor with high refractive index response sensitivity was developed. By fabricating a waist-enlarged bitaper structure on the interference arm of a single mode–multimode–single mode (SMS) Mach–Zehnder interferometer (MZI), the spectral contrast and response sensitivity were improved. Subsequently, the response sensitivity was further improved by etching the interference arm. When a beam of light was introduced into the sensor, due to the structural mismatch between the multimode fiber and the normal transmission light, the difference between the low-order mode and the high-order mode was generated in the fiber core and the fiber cladding. In the process of transmission in the sensing arm, due to the different refractive indices of the core and cladding, the optical path difference of the high-order mode and the low-order mode was different, which eventually generated interference fringes. The experimentally measured response sensitivity of SMS MZI in the range of 1.351 RIU to 1.402 RIU is 57.623 nm/RIU; the response sensitivity of a single mode–multimode–bitaper–multimode–single mode (SMBMS) MZI is 61.607 nm/RIU; and the response sensitivity of the etched SMBMS (ESMBMS) MZI is 287.65 nm/RIU. The response sensitivity of the new ESMBMS MZI is three times higher than that of the original SMS MZI. The sensor has the characteristics of compact structure, high sensitivity, easy manufacture, and a wide range of refractive index measurements, and can be used in food processing, pharmaceutical manufacturing and other fields.

## 1. Introduction

The refractive index is a commonly used process control index in food production, pharmaceutical development, and other fields. By measuring the refractive index of liquid substances, the composition of the substance can be identified, the concentration determined, and the degree of purity and quality judged. Traditional electronic sensors cannot work in harsh environments, such as high salinity and strong oxidation [1,2]. In recent years, optical fiber sensors had the advantages of excellent anti-electromagnetic interference, good information security, and high precision. The wavelength-dependent type [3] is different from the energy-dependent type [4], which has the advantages of high precision and no interference from light source power.

Optical fiber wavelength-dependent sensors are divided into various types according to their structure, such as the fiber Michelson [5], U-shaped fiber sensor [6,7], coated fiber sensor [8], and Mach–Zehnder interferometer (MZI) [7,8,9,10,11,12,13,14,15,16], etc. In 2019, Wang et al. [5] developed a Michelson interferometer by splicing a single-mode fiber and a hollow quartz tube, based on a phase demodulation method, in the range of 1.331 RIU to 1.387 RIU; the refractive index response sensitivity is 8.1498 rad/RIU. The proposed sensor was compact and low-cost, but demodulation analysis was difficult, due to the Fourier analysis of the measurement results. In the same year, Danny et al. [6] proposed a theoretical model of a U-shaped fiber probe, and used the ray tracing method to realize refractive index sensing. In 2020, Wang et al. [7] fabricated a U-shaped double-side polished optical fiber refractive index sensor, with a refractive index response sensitivity of 1541%/RIU from 1.33 RIU to 1.39 RIU. The design and development of the U-shape structure still had certain theoretical and manufacturing difficulties, which needed to be further improved. In 2020, Diegueza et al. [8] reported an optical fiber refractive index sensor coat, with a thin copper film to increase the contrast of the interference fringes, in 0 to 18% glycerol solutions; the response sensitivity is 19 pm/(Glycerol% by weight). Since the development of optical fiber sensing that requires technical support, such as coating, there were certain limitations in production and cost. MZI is small in size, high in sensitivity, and simple in production. Currently, MZI is a hot topic in various research groups.

Since the 1990s, the optical fiber MZI, used to measure the refractive index, began to develop rapidly as a new generation of sensors. In 1994, Liang [9] introduced how to use MZI to measure the refractive index of air. Compared with other methods using double-beam interferometry, it was characterized by convenient operation, stable and reliable data, and easy resolution. In their study of 2014, An et al. [10] measure a humidity response of 0.223 nm/%RH from 35% RH to 85% RH, based on a MZI coated with polyvinyl alcohol material.

In order to further improve the response sensitivity of the sensor, many researchers explored methods to change the structure of the MZI sensing arm, such as adding a fiber taper structure. In 2019, Liao et al. [11] designed an optical fiber MZI based on the fiber taper and bubble structure for ethanol concentration measurement, and record a sensitivity of 28 nm/vol from 0.3 vol to 0.7 vol. In 2019, Vahid et al. [12] fabricated a MZI with ultra-thin sensor arms based on a custom flame-based tapering machine. The sensor’s cladding diameter is only 35.5 µm, and the refractive index sensitivity is 415 nm/RIU from 1.332 RIU to 1.384 RIU. In 2019, based on the cascaded up-down taper, Han et al. [13] sandwiched a polarization-maintaining fiber between two single-mode fibers, and they record a refractive index sensitivity of −310.40 dB/RIU from 1.3164 RIU to 1.3444 RIU.

Etching the MZI sensing arm is also a commonly used method in improving the refractive index sensitivity. In 2011, Changping Tang [14] developed a MZI formed by splicing a section of solid-core photonic crystal fiber between two sections of single-mode fiber, and the sensitivity is 70.45 nm/RIU from 1.340 RIU to 1.384 RIU. Then, the coupling degree between the sensor interference light field and the external refractive index is further improved by corrosion, and the sensitivity increases to 198.77 nm/RIU, which is about 2.8 times than before corrosion. In 2019, Huang et al. [15] fabricated a MZI by splicing a photonic crystal fiber between two single-mode fibers, and placed the sensing arm in 40% hydrofluoric acid to reduce the cladding thickness. The experimental results show that the sensitivity increases almost three-fold. In 2019, Haifeng et al. [16] inserted a photonic crystal fiber between two single-mode fibers, and record a sensitivity of 106.19 nm/RIU from 1.333 RIU to 1.381 RIU. After etching the sensor, the cladding diameter is reduced from 250 μm to 112 μm, and the sensitivity improves to 211.53 nm/RIU. In conclusion, an increase in the sensitivity of the refractive index response can be achieved by structural change and cladding etching.

In this paper, based on the principle of double-beam interference, three sensors were designed, fabricated, and compared. The first was a fiber sensor, based on single mode–multimode–single mode (SMS) MZI. The second specific structure was realized by melting a waist-enlarged bitaper in the middle of the sensing arm of the first sensor; the specific structure was single mode–multimode–bitaper–multimode–single mode (SMBMS). The third was realized by etching the sensing arm with hydrofluoric acid on the basis of the second sensing structure, which was referred to as ESMBMS for short. The principle of interference sensing, the fabrication method of the optical fiber cone structure, the analysis of the spectral mode, and the corrosion mechanism of the optical fiber by hydrofluoric acid were analyzed; experiments were designed to measure the sensitivity response of the refractive index of various sensors. Finally, a high-sensitivity refractive index sensor was obtained by making waist-enlarged bitaper and etching. For the convenience of comparison, we summarize the characteristics, advantages, and disadvantages of various sensing structures in Table 1. The comparison shows that the sensor developed in this paper has the advantages of low price, high sensitivity, being simple to make, easy to read, and so on, which has high practical value in the fields of food processing and pharmaceutical production.

## 2. Principle and Fabrication

### 2.1. Single mode–Multimode–Single mode (SMS) Mach–Zehnder Interferometer (MZI)

#### 2.1.1. Structural Design

The basic structure of the fiber MZI involved in this paper is SMS MZI, as shown in Figure 1. This MZI included the following parts: the left side was the single mode fiber of the input light, the middle section was a 20 mm long multimode fiber, and the right side was the single mode fiber of the output light. The working principle of MZI is as follows: When light is transmitted from a single mode fiber into a multimode fiber, a part of the cladding mode is excited. Due to the different refractive indices of the core and cladding, there is a certain optical path difference between the core mode and the cladding mode. When light re-entered the single mode fiber from the multimode fiber, the core mode and the cladding mode were coupled and interfered to form interference fringes.

#### 2.1.2. Principles of Refractive Index Sensing

The occurrence of the interference phenomenon depends on the optical path difference between the two beams. It can be seen from the structure that the core path and cladding path of the MZI sensing arm are of equal length. There is a direct relationship to the frequency spectrum of wavelengths, which is the effective refractive index difference between the different light wave modes. Assuming two different guided modes *LP*_0*m*_ and *LP*_0*n*_ (*m*, *n* are positive integers) are in the multimode fiber, the phase difference between them is:(1)$$\Delta \varphi^{}=\frac{2\pi (n_{eff}^{m}-n_{eff}^{n})L}{\lambda }=\frac{2\pi \Delta n_{eff}^{m,n}L}{\lambda }$$
where *L* is the length of the fiber and λ is the input wavelength of the light source. $\Delta n_{eff}^{m,n}$ is the effective refractive index difference between *LP*_0m_ and *LP*_0n_. The equation for the effective refractive index is as follows:(2)$$n_{eff}=\frac{n\cdot s}{L}$$
where *n*_eff_ is the effective refractive index, *n* is the refractive index in the medium, *s* is the distance traveled by the light, and *L* is the length of the interference arm. Equation (2) shows that when the refractive index of the medium and *L* are constant, the mode with the higher order has a larger diffusion angle. That is, the effective refractive index of the high-order mode is greater than that of the low-order mode.

The intensity of the output light can be expressed as:(3)$$I=I_{1}+I_{2}+2\sqrt{I_{1}I_{2}}\cos\Delta \varphi_{}^{}$$
where *I* is the output light intensity of the MZI, *I_1_* and *I_2_* are the light intensity of the guided mode *LP*_0*m*,_ and the light intensity of *LP*_0*n*_ in the interference core. According to Equation (3), the phase difference equation corresponding to the valley is as follows:(4)Δφ=(2m+1)π

Substituting Equation (1) into Equation (4), we obtain:(5)$$\frac{2\pi \Delta n_{eff}^{m,n}L}{\lambda_{m}}=(2m+1)\pi$$
where *m* is an integer, representing the interference order and $\lambda_{m}$ is the center wavelength of m-order interference.

According to Equations (4) and (5), Equation (6) is obtained:(6)$$\delta \lambda_{m}\approx 2\pi L\delta n_{eff}$$
where $\delta \lambda_{m}$ is the center wavelength shift of the m-th order interference fringes and $\delta n_{eff}$ is the change caused by the refractive index of the sucrose solution. Equation (6) demonstrates that when the interference length *L* is constant, the shift amount of the interference valley wavelength changes with the change of the refractive index of the external liquid. Therefore, the refractive index of the sucrose solution is measured by monitoring the shift in the wavelength of the *m*-th valley of the MZI.

### 2.2. Single mode–Multimode–Bitaper–Multimode–Single mode Mach–Zehnder Interferometer (SMBMS MZI)

#### 2.2.1. Structural Design

In order to enhance the sensitivity of the fiber MZI, the SMBMS MZI was developed on the basis of the SMS MZI, as shown in Figure 2; a waist-enlarged bitaper was fused to the middle multimode segment. The function of the fusion point in front of the waist-enlarged bitaper was to distribute the light transmitted through the single mode fiber to the multimode fiber core and fiber cladding, while the function of the waist-enlarged bitaper was to redistribute the light transmitted in the core and the cladding, which excited higher-order modes and entered the multimode cladding transmission section behind the waist-enlarged bitaper. The function of the fusion point behind the waist-enlarged bitaper was to couple the light transmitted in the core and the cladding to the output fiber, and through the spectrometer for storage and analysis. In order to verify whether the position of the waist-enlarged bitaper had an effect on the sensor performance, we made a control sensor, and set the position of the waist-enlarged bitaper at one-third of the sensing arm, as shown in Figure 3.

#### 2.2.2. Analysis of Coupling Response Characteristics of Waist-Enlarged Bitaper

The following section focused on analyzing the response characteristics of the optical fiber waist-enlarged bitaper. The inverted taper was composed of a multimode fiber and an inverted taper fiber with a gradually enlarged diameter. The waist-enlarged bitaper part played the role of expanding the beam of the fundamental mode spot. The waist-enlarged bitaper directly affected the propagation response characteristics of the beam in it, as shown in Figure 4. Therefore, we used the waist-enlarged bitaper to make more light waves leak into the cladding and excite higher-order modes. In the following, by explaining the propagation response characteristics of Gaussian beams in tapered fibers, the reasons for the improvement of the coupling efficiency by the waist-enlarged bitaper are theoretically given.

The propagation response characteristics of Gaussian beams in tapered fibers are given by the spot size *ω(z)* and the wavefront curvature radius *R(z),* used to describe the propagation response characteristics of Gaussian beams [17].
(7)$$\omega^{2}(z)=\omega^{2}(0)\cdot {\left[A^{2}+{\left(B/k\omega^{2}(0)\right)}^{2}\right]}^{\frac{1}{2}}$$
(8)$$\frac{1}{R(z)}=\frac{\omega^{\prime }(z)}{\omega (z)}$$

First, the above two equations can change to the following form:(9)$$\omega^{2}(z)=\omega^{2}(0)[(g(0)/g(z))\cos^{2}\theta +\frac{1}{g(0)g(z)}\cdot \frac{\sin\theta^{2}}{k_{1}^{2}\omega (0)}]$$

This can be obtained by derivation:(10)$$\omega^{2}(z)=a_{0}f(z)/(\frac{2\pi n_{1}}{\lambda })\sqrt{2\Delta }$$

The spot magnification factor M can be expressed as:(11)M=ω(z)/ω(0)=f(z)=1+c⋅expγ(z)

Therefore, at the position of the waist-enlarged bitaper, the light spot was enlarged. In a coupled system, the coupling tolerance of the system was only related to the spot size. The larger the spot size, the larger the vertical and horizontal tolerances. In a word, the waist-enlarged bitaper played the role of spot amplification in the optical fiber coupling system, improving the coupling tolerance of the system, and redistributing the spectrum.

#### 2.2.3. Manufacture

Fiber MZI waist-enlarged bitaper is manufactured using a common fusion splicer (FITEL, S178A). In the sensing system, a fiber cleaver (Furukawa, S325) was used to obtain a flat fiber end, and make a waist-enlarged bitaper based on commercial fusion splicers. Multimode optical fiber has a core diameter of 40 µm and a cladding diameter of 125 µm. By increasing the splicing time to 500 ms, and increasing the splicing strength to 155 unit, the waist-enlarged bitaper is obtained. Figure 5 is the photo of the waist-enlarged bitaper observed through a microscope. The diameter of the waist-enlarged bitaper is expanded to 152 μm, and the length of the cone region is 350 μm.

### 2.3. Etched Single mode–Multimode–Bitaper–Multimode–Single mode Mach–Zehnder Interferometer (ESMBMS MZI)

#### 2.3.1. Structural Design

Since the refractive index sensitivity response of the SMBMS type structure sensor was still very low, in order to further improve the refractive index sensitivity, the SMBMS type structure sensor was improved. As shown in Figure 6 and Figure 7, the structure of the ESMBMS is based on the structure of the SMBMS, and the middle multimode segment is etched in hydrofluoric acid.

When the sensing arm was etched, the diameter of the interference arm decreased, which increased the coupling degree between the cladding mode and the ambient refractive index. The change of the refractive index caused the change of the phase difference of the transmitted light in the fiber, and then the change of the refractive index was measured by measuring the change of the interference spectrum. In this experiment, the sensing arm was used to sense the change of refractive index, which was reflected on the spectrum, that is, the wave valley moved. By measuring the drift of the valley, the change in the refractive index was measured.

#### 2.3.2. The Principle of Chemical Corrosion

The etching solution often used in the optical fiber chemical etching method is a hydrofluoric acid solution. The cladding of optical fibers is generally made of silicon dioxide (SiO_2_) material. Hydrofluoric acid dissolves silica mainly for the following two reasons: First, hydrofluoric acid solution contains fluoride ions, which combine with silicon ions in SiO_2_ to form complex ions. Second, the hydrogen ions in the hydrofluoric acid solution form water with the oxygen ions in the SiO_2_. The reaction equation can be expressed as follows:(12)$${\mathrm{SiO}}_{2}+6\mathrm{HF}=2H_{3}O{}^{+}+{\mathrm{SiF}}_{6}^{2-}$$

In the experiment, we first fixed the sensing unit of the interferometer in a plastic dish, and connected the broadband light source and the spectrometer at both ends. According to past experience, the corrosion rate of 40% hydrofluoric acid at 23 °C is about 2 μm/min [18,19]. After etching the MZI for 20 min, the corrosion surface was relatively smooth.

#### 2.3.3. Manufacture

Figure 5 shows the MZI waist-enlarged bitaper before etching, and Figure 8 shows the bitaper after etching. The 40% hydrofluoric acid solution was directly dropped on the optical fiber sensing arm and etched for 20 min. The diameter of the sensing arm of the 40/125 multimode fiber is reduced to 85 μm. The diameter of the fiber bitaper is reduced from 152 μm to 112 μm, and the length of the waist-enlarged bitaper shows little change.

## 3. Spectral Pattern Analysis

There may be multiple modes in the fiber, and due to the different propagation constants between these modes, the optical path difference between the modes will appear under the same transmission length. When these modes are coupled, interference between the modes occurs. Through fabrication experiments, it is found that the position of the waist-enlarged bitaper has little effect on the SMBMS and ESMBMS spectra, so the position of the waist-enlarged bitaper is not distinguished when displaying the spectra. Figure 9 shows the transmission spectra of the MZIs, which show that the density of the interference fringes change due to the addition of waist-enlarged bitaper and corrosion.

According to the transmission spectrum, the higher-order mode is found, and the frequency spectrum obtained after Fourier transform is shown in Figure 10. The main peak amplitudes are located at 0.0199164 nm^−1^, 0.0300104 nm^−1^, and 0.0412538 nm^−1^.

Using the Taylor expansion to expand the wavelength the phase ϕ is formulated as:(13)$$\phi \approx \phi_{0}-\frac{2\pi \Delta \lambda }{\lambda^{2}}\Delta n_{eff}\cdot L$$
where Δλ is the wavelength difference, $\phi_{0}$ is a initial phase, and λ is the wavelength of spectral valley.

Due to the MZIs spectra corresponding to mathematical cosine patterns, the following equation is obtained:(14)cosΔϕ=cos(2πξΔλ)

If $\phi_{0}$ is the initial phase, and we assume it is equal to 0, the spatial frequency *ξ* [20] is:(15)$$\xi =\frac{1}{\lambda^{2}}\Delta n_{eff}\cdot L$$

Considering the modal dispersion, we established the relationship between Δn_eff_ and different modes, based on OptiFiber. The wavelength λ is around 1550 nm, and the lengths of the MZIs are all 20 mm. Through the Fourier transform, the spatial frequency *ξ* are 0.0199164 nm^−1^, 0.0300104 nm^−1^, and 0.0412538 nm^−1^. Therefore, the parameter Δn_eff_ calculated from Equation (15) are 0.002392458, 0.003604999, and 0.004955613. In other words, as the sensing arm is spliced with waist-enlarged bitaper and etched, higher-order modes are excited, affecting the optical path of transmitted light in the optical fiber core and cladding. Subsequently, the refractive index responses of the three sensors were measured experimentally.

## 4. Experiment

### 4.1. The Refractive Index Sensitivity Response Characteristics of SMS MZI

Figure 11 is a schematic diagram of a refractive index sensing system. The measurement system consisted of a broadband light source (BBS, Lightcomm, Shenzhen, China, ASE-CL), a spectrum analyzer (OSA, Anritsu, Kitakyushu, Japan, MS9740A), and a vessel for placing the refractive index solution. The spectrometer resolution was set to 0.02 nm, and the bandwidth of the BBS was 80 nm. In the experiment, sucrose solutions with different concentrations were used as refractive index samples, and their refractive indices are 1.351, 1.369, 1.379, 1.387, 1.394, and 1.402 after being tested by Abbe’s refractive index detector. The MZI was uniformly soaked in a sucrose solution of each refractive index for 5 min at a stable room temperature of 23 °C. The wavelength shift was observed and the data were recorded.

The length of the sensing arm of the SMS MZI is about 20 mm. From Figure 12, we see that there is only one wavelength valley in the spectrum of the SMS MZI. We monitored the wavelength of the valley, and Figure 13 shows that the refractive index sensitivity of SMS MZI is 57.623 nm/RIU, and the linearity of the refractive index sensitivity response characteristic is 0.999. Therefore, as the refractive index sensitivity is a positive number, it is seen that the wavelength shifts to the long-wavelength direction; however, the refractive index sensitivity of SMS MZI is too low.

### 4.2. The Refractive Index Sensitivity Response Characteristics of SMBMS MZI

In view of the low refractive index sensitivity of SMS MZI, an improvement was made on the basis of SMS MZI, by melting a waist-enlarged bitaper in the middle of the sensing arm. The photo of the waist-enlarged bitaper is shown in Figure 5. The SMBMS MZI was fabricated, in which the length of the sensing arm of the SMBMS sensor is about 20 mm, the experimental temperature is 23 °C, and the refractive index range is from 1.351 RIU to 1.402 RIU. The sensing arms were uniformly soaked in a sucrose solution for each refractive index for 5 min. The wavelength drift was observed and the data were recorded, shown in Figure 14. At the same time, we changed the position of the waist-enlarged bitaper, and found that the position of the waist-enlarged bitaper had little effect on the spectral shape. Subsequently, we performed refractive index experiments on the sensor with the waist-enlarged bitaper position at one-third of the sensing arm, shown in Figure 15.

From Figure 14 and Figure 15, we see that there are two valleys within the spectrum of SMBMS MZI. In order to facilitate the comparison with SMS MZI, the valley of the wavelength around 1536 nm was selected for monitoring. Figure 16 shows that the refractive index sensitivity of the SMBMS MZI is 61.607 nm/RIU, and the linearity of the refractive index sensitivity response is 0.999. It can be seen that the wavelength still drifts in the long-wavelength direction, and the refractive index sensitivity of SMBMS MZI improves, compared to SMS MZI. Overall, the sensitivity improvement effect is still not obvious. The reason for this is that the diameter of the sensing arm was relatively thick, and the coupling effect with the refractive index solution was not obvious.

### 4.3. The Refractive Index Sensitivity Response Characteristics of ESMBMS MZI

Since the refractive index sensitivity of SMBMS MZI was still low, improvements were made on the basis of SMBMS MZI. The method used was to corrode the sensing arm of SMBMS MZI with hydrofluoric acid for 20 min. The waist-enlarged bitaper is shown in Figure 8. The sensing arm of the fabricated ESMBMS MZI is about 20 mm long, and the fiber diameter is reduced by 40 μm after 20 min of hydrofluoric acid etching at 23 °C. The refractive index of sucrose solution ranges from 1.351 RIU to 1.402 RIU. The sensing arm was immersed in the sucrose solution of each refractive index for 5 min to observe the wavelength and record the data, shown in Figure 17 and Figure 18.

It can be seen from Figure 17 and Figure 18 that for monitoring the valley with the center wavelength around 1545 nm, the refractive index sensitivity of ESMBMS MZIs is 287.65 nm/RIU, and the linearity of the refractive index sensitivity response is 0.999, as shown in Figure 19. Therefore, it is demonstrated that the wavelength still drifts to the long-wavelength direction, and the refractive index sensitivity of ESMBMS MZI increases almost four-fold compared to SMBMS MZI. Overall, the advantages of ESMBMS MZI refractive index sensitivity response characteristics are obvious. The reason is that the sensing arm cladding of the SMBMS structure was etched with hydrofluoric acid to reduce its diameter, which greatly improved the sensitive response characteristics. Therefore, the refractive index sensitivity of ESMBMS MZI is obviously improved. 

## 5. Conclusions

In this paper, a high-sensitivity MZI was designed and fabricated. The single mode fiber and the multimode fiber were directly spliced by a commercial fusion splicer to form a coupling point. The multimode fiber was in the middle of the sensing arm, and a waist-enlarged bitaper was fused in the middle of the sensing arm, used to realize beam adjustment. Finally, the cladding of the sensing arm was etched with hydrofluoric acid to reduce its diameter, which in turn achieved improved response sensitivity. Fourier transform analysis shows that both the waist-enlarged bitaper and erosion excite higher-order modes, resulting in a larger optical path difference for light transmitted in the optical fiber core and cladding. The refractive index sensitivity responses of these three sensors were experimentally investigated, and the refractive index sensitivity of SMS MZI is 57.623 nm/RIU, with a linearity of 0.9795; the refractive index sensitivity of SMBMS MZI is 61.607 nm/RIU, with a linearity of 0.9545; and the refractive index sensitivity of ESMBMS MZI is 287.65 nm/RIU, with a linearity of 0.9843. It is shown that ESMBMS MZI has a very high refractive index sensitivity and good linearity. As the new ESMBMS MZI, designed in this paper, possesses the advantages of high sensitivity, good linearity, low cost, and simple fabrication, it has a high practical value and good application prospects for future production.

## Figures and Tables

**Figure 1 micromachines-13-00689-f001:**
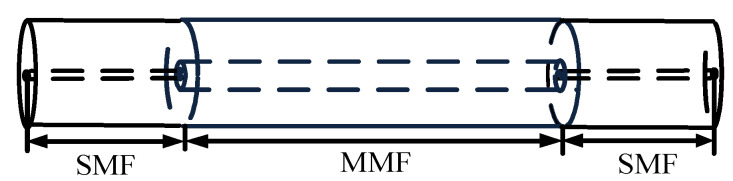
Schematic diagram of the Single mode–Multimode–Single mode Mach–Zehnder Interferometer (SMS MZI).

**Figure 2 micromachines-13-00689-f002:**
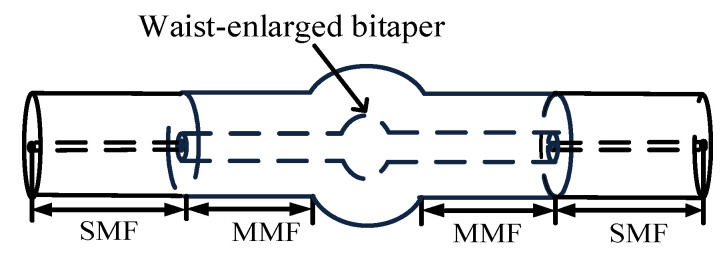
Schematic diagram of the Single mode–Multimode–Bitaper–Multimode–Single mode Mach–Zehnder Interferometer (SMBMS MZI).

**Figure 3 micromachines-13-00689-f003:**
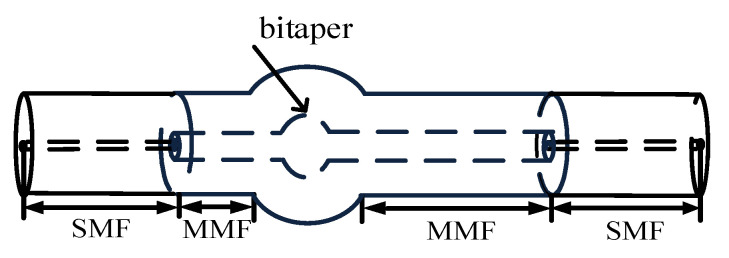
Schematic diagram of the SMBMS MZI with the position of the bitaper set to one third of the sensing arm.

**Figure 4 micromachines-13-00689-f004:**
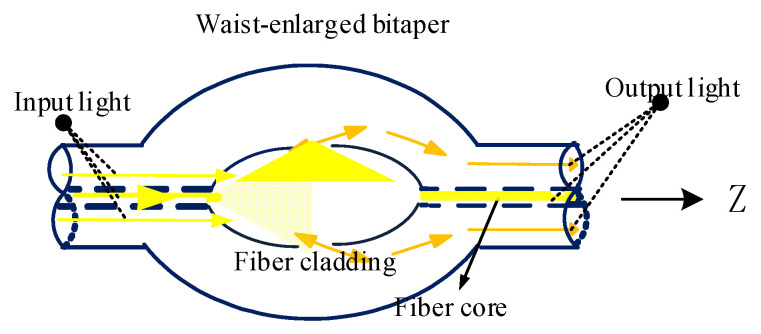
Optical path analysis of waist-enlarged bitaper in SMBMS MZI.

**Figure 5 micromachines-13-00689-f005:**
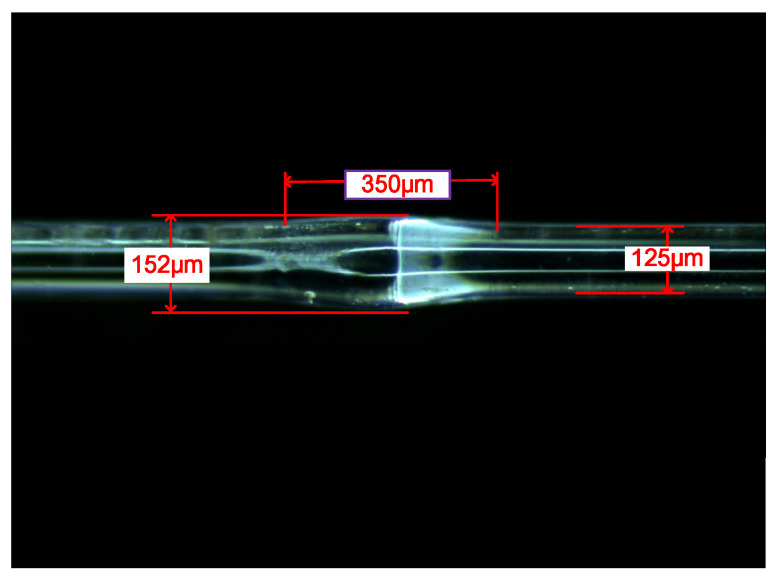
The waist-enlarged bitaper in SMBMS MZI.

**Figure 6 micromachines-13-00689-f006:**
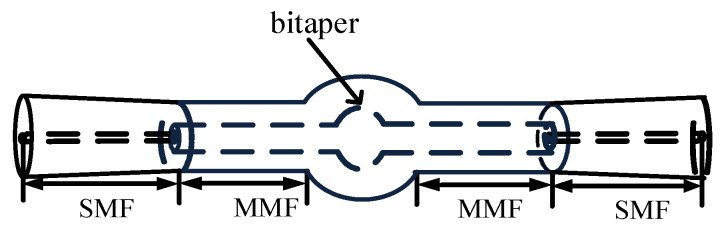
Schematic diagram of the Etched Single mode–Multimode–Bitaper–Multimode–Single mode Mach–Zehnder Interferometer (ESMBMS MZI).

**Figure 7 micromachines-13-00689-f007:**
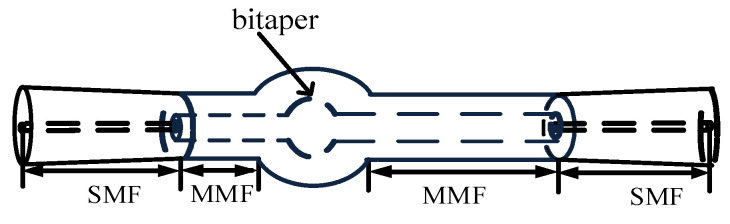
Schematic diagram of the ESMBMS MZI with the position of the bitaper set to one third of the sensing arm.

**Figure 8 micromachines-13-00689-f008:**
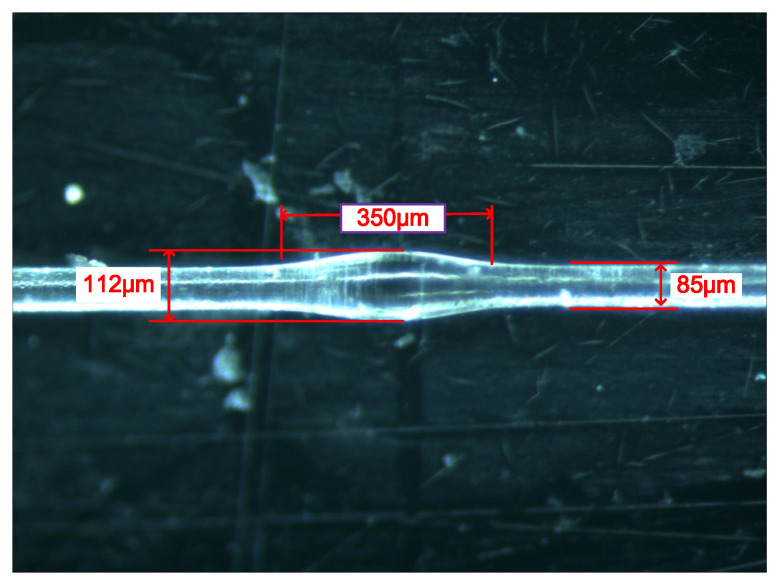
The etched waist-enlarged bitaper in ESMBMS MZI.

**Figure 9 micromachines-13-00689-f009:**
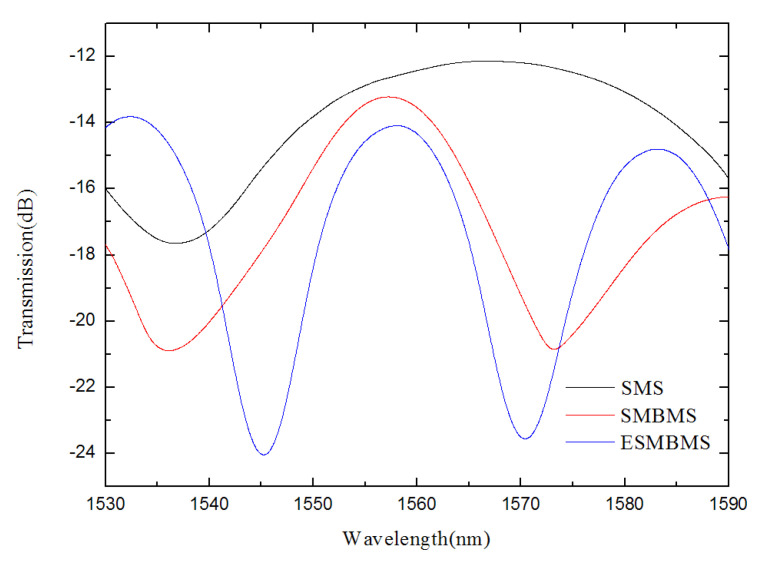
Measured transmission spectra with different L.

**Figure 10 micromachines-13-00689-f010:**
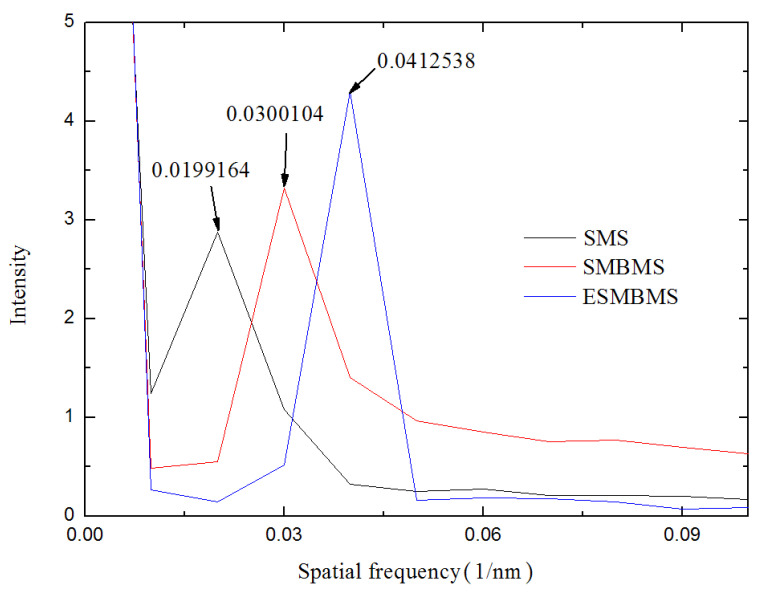
Spatial frequency spectra of the MZIs with different lengths.

**Figure 11 micromachines-13-00689-f011:**
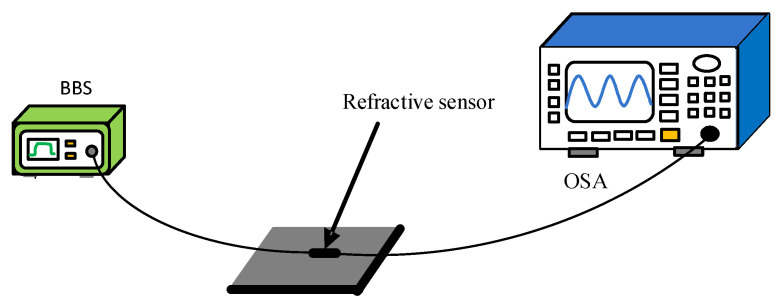
Refractive index sensing experimental device schematic diagram.

**Figure 12 micromachines-13-00689-f012:**
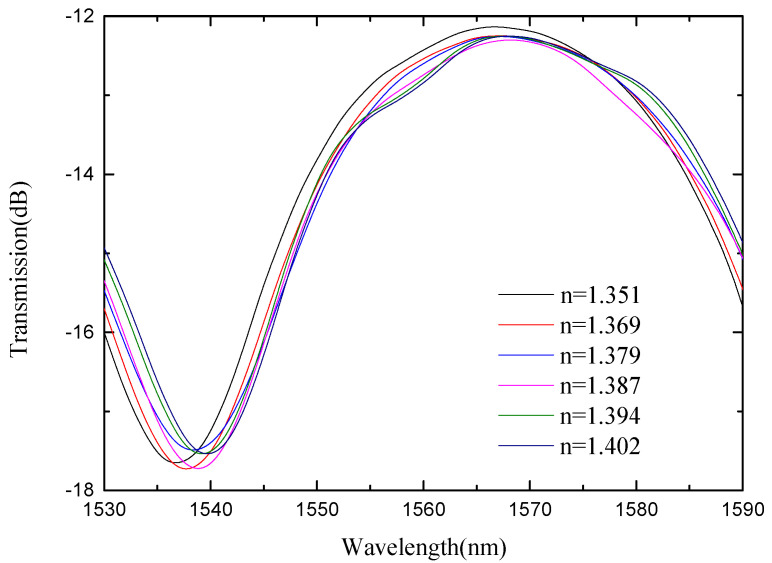
Spectra of SMS MZI at different refractive indices.

**Figure 13 micromachines-13-00689-f013:**
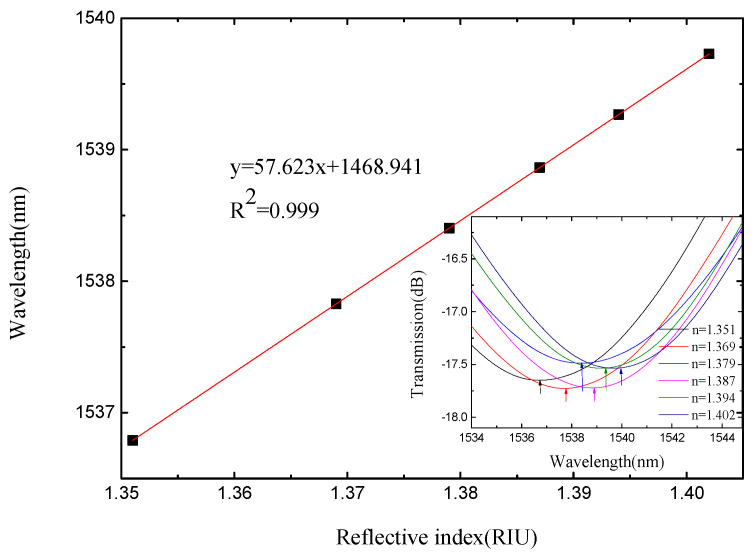
The sensitivity response characteristic diagram of SMS MZI.

**Figure 14 micromachines-13-00689-f014:**
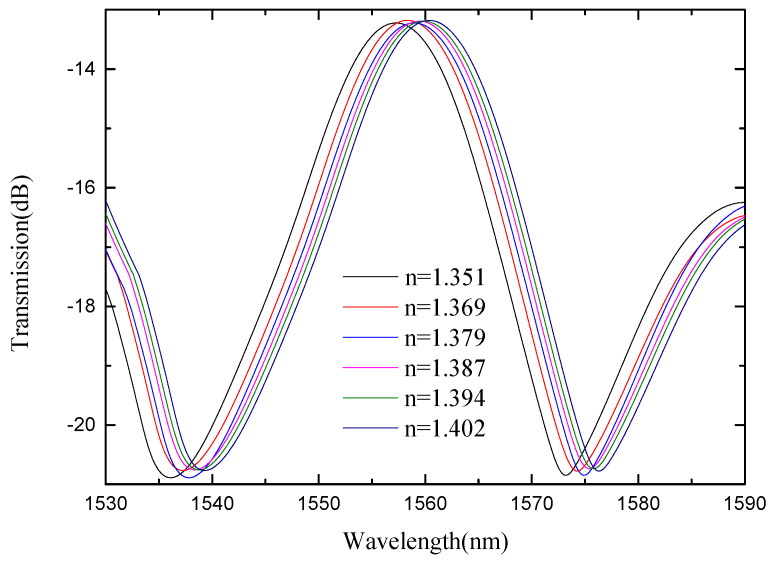
Spectra of SMBMS MZI(1/2L) at different refractive indices.

**Figure 15 micromachines-13-00689-f015:**
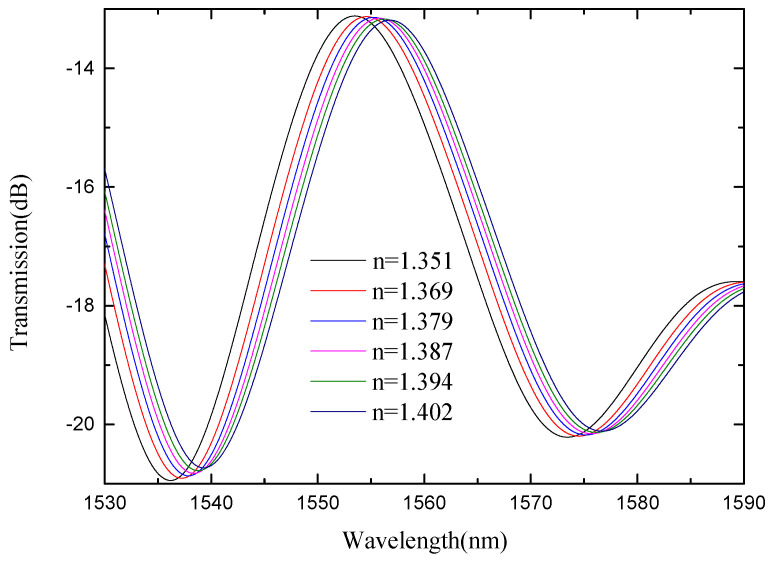
Spectra of SMBMS MZI(1/3L) with the position of the bitaper set to one third of the sensing arm at different refractive indices.

**Figure 16 micromachines-13-00689-f016:**
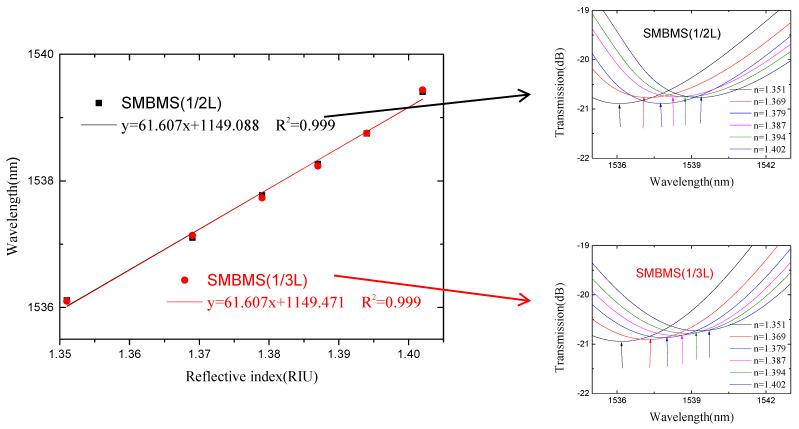
The sensitivity response characteristic diagram of SMBMS MZI.

**Figure 17 micromachines-13-00689-f017:**
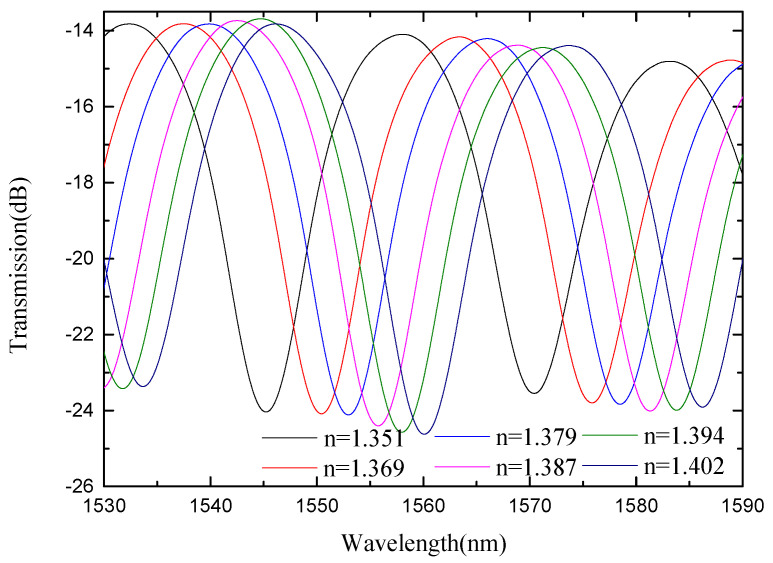
Spectra of ESMBMS MZI(1/2L) at different refractive indices.

**Figure 18 micromachines-13-00689-f018:**
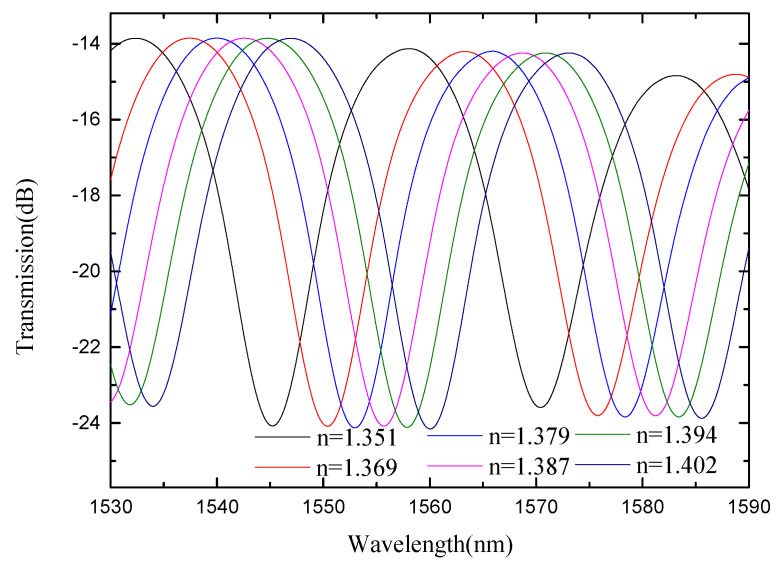
Spectra of ESMBMS MZI(1/3L) with the position of the bitaper set to one third of the sensing arm at different refractive indices.

**Figure 19 micromachines-13-00689-f019:**
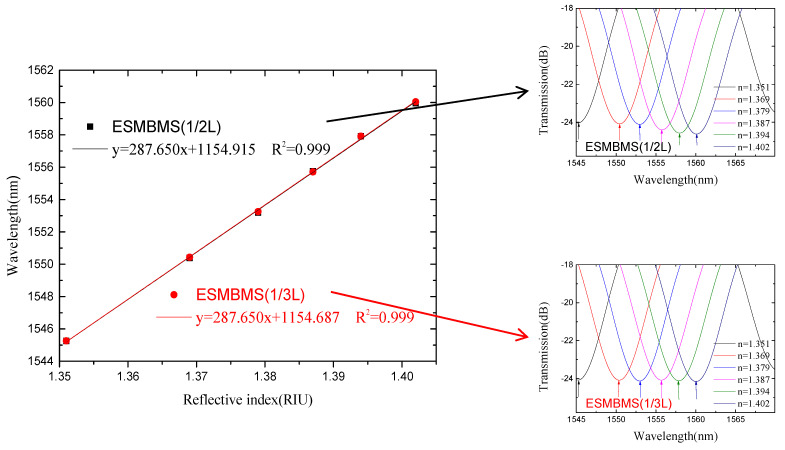
The sensitivity response characteristic diagram of ESMBMS MZI.

**Table 1 micromachines-13-00689-t001:** Comparison of various refractive index sensors.

Structure Type	Monitoring Volume	Range	Sensitivity	Advantages	Insufficient	Ref
Michelson probe	Pattern	1.331RIU to 1.387RIU	8.1498 rad/RIU	Compact and low-cost	The demodulation analysis was difficult	[5]
U-shape fiber probe	Ray tracing method	1.33RIU to 1.39RIU	1541%/RIU	Flexible structural design	Theoretical and manufacturing difficulties, poor repeatability	[6,7]
Optical fiber sensor coat with a thin copper film	Wavelength	0 to 18% glycerol solutions	19 pm/(Glycerol % by weight)	High response sensitivity	The process of optical fiber coating is complex	[8]
MZI coated with polyvinyl alcohol material	Wavelength	35% RH to 85%RH	0.223 nm/%RH	High response sensitivity	The process of optical fiber coating is complex	[10]
MZI based on the fiber taper and bubble structure	Wavelength	0.3 vol to 0.7 vol	28 nm/vol	Wavelength type measurement is not affected by light source, connector, etc.	Low response sensitivity	[11]
MZI with ultra-thin sensor arms	Wavelength	1.332RIU to 1.384RIU	415 nm/RIU	High response sensitivity	Ultra-thin fiber is expensive	[12]
MZI based on polarization-maintaining fiber	Strength	1.3164RIU to 1.3444RIU	310.40 dB/RIU	The price of demodulation equipment is very low	Polarization- maintaining fiber is expensive	[13]
MZI based solid-core photonic crystal fiber	Wavelength	1.340RIU to 1.384RIU	70.45 nm/RIU	Wavelength type measurement is not affected by light source, connector, etc.	Solid-core photonic crystal fiber is expensive	[14]
MZI based photonic crystal fiber	Wavelength	1.333RIU to 1.381RIU	211.53 nm/RIU	Sensitivity can be effectively improved by etching	Photonic crystal fiber is expensive	[16]
ESMBMS MZI	Wavelength	1.351RIU to 1.402RIU	287.65 nm/RIU	Low price, high sensitivity, simple to make, easy to read	The demodulation equipment has not been independently developed.	This paper

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
