# Peer review of "High Sensitivity Optical Fiber Mach–Zehnder Refractive Index Sensor Based on Waist-Enlarged Bitaper"

_micromachines, 2022, doi:10.3390/mi13050689_

Round 1

Reviewer 1 Report

The authors performed sensing experiments based on three different MZI structures, i.e., SMS MZI, SMBMS, and ESMBMS, showing an increased sensitivity within the same RI range. This manuscript is completed with theoretical analysis and experimental results. However, poor writing, grammar errors, obvious mistakes, etc., are somehow obstacles to the readability of this manuscript. In addition, it lacks adequate comparison and discussion with schemes in the literature, which will show the merits or advantages of their sensor more clearly. Insight of this, rejection with resubmission is recommended for this paper in my view.

The detailed comments can be seen below:

  1. In the section Introduction, the text tense is suggested by using past tense;
  2. “Jin Wang et al. [5] of Nanjing information technology university develop”is suggested to revise as “ Wang et al. Developed...”. Similar suggestion for the following context;
  3. “U-shaped structure” in line 55 can be “U-shape structure”;
  4. All the figures are better to be centered;
  5. “formula”in this manuscript should be “equation”;
  6. “By increasing the splicing time to 500s”in line 203, 500s should be 500ms?
  7. The authors stated that “it can be seen from figure 11 that the refractive index sensitivity of SMS MZI is 57.623 nm/RIU, and the linearity of the refractive index sensitivity response characteristic is 0.9795.”However, the linearity in Figure 11 shows 0.999! The same issue can be seen in Figure 13;
  8. Figure 12 seems not completed with some curves missing;
  9. “The refractive index of sucrose solution ranges from 1.351 RIU to 1.4067 RIU.”However, Figure 14 shows the maximum RI of 402 instead of 1.4067;
  10. The format of the reference seems chaoticand needsto be modified;
  11. I noticed some schemes in the Introduction showeven better sensitivity thanthe advantages of the sensor proposed by the authors. It is better to have a discussion or comparison to clear the merits and purpose of this manuscript.

Author Response

Dear reviewer:

Thank you very much for your comments about our paper submitted to High sensitivity optical fiber Mach-Zehnder refractive index sensor based on waist-enlarged bitaper (micromachines-1703654). After carefully studying the comments and your advice, we have made corresponding changes.

The following is the answers and revisions we have made in response to the reviewer's questions and suggestions item by item.

Reviewer #1

  1. Comments:In the section Introduction, the text tense is suggested by using past tense;

Revision: Thanks for your reminder, we have changed the tense of the introduction to the past tense.

    The location of the revision: Introduction of the second, third, fourth, fifth and sixth paragraphs.

  1. Comments:“Jin Wang et al. [5] of Nanjing information technology university develop”is suggested to revise as “ Wang et al. Developed...”. Similar suggestion for the following context;

Revision: Thanks for your suggestion, we have made changes to the corresponding section in the context, and the specific location is as follows.

    The location of the revision: Introduction of the second, third, fourth and fifth paragraphs

  1. Comments:“U-shaped structure” in line 55 can be “U-shape structure”;

Revision: We have changed "U-shaped structure" to "U-shape structure".

    The location of the revision: The thirteen line of the second paragraph in the introduction.

  1. Comments:All the figures are better to be centered;

Revision: Thanks for your suggestion, we've settled the pictures in the middle.

    The location of the revision: All the pictures in the manuscript.

  1. 5. Comments:“formula”in this manuscript should be “equation”;

Revision: We have changed "formula" to "equation" in the manuscript.

  1. 6. Comments:“By increasing the splicing time to 500s”in line 203, 500s should be 500ms?

Revision: We are very sorry for our typo, and have changed 500 s to 500 ms in the manuscript.

    The location of the revision: The thirteen line of the second paragraph in the 2.2.3 Manufacture.

  1. 7. Comments:The authors stated that “it can be seen from figure 11 that the refractive index sensitivity of SMS MZI is 57.623 nm/RIU, and the linearity of the refractive index sensitivity response characteristic is 0.9795.”However, the linearity in Figure 11 shows 0.999! The same issue can be seen in Figure 13;

Revision: We are very grateful to the reviewers for pointing out our mistakes. We have corrected the mistakes and checked the whole manuscript, and corrected the wrong data in Figure 11 and Figure 13 of the original manuscript.

    The location of the revision: The fifth line above figure 12. The fifth line below figure 16.

  1. 8. Comments:Figure 12 seems not completed with some curves missing;

Revision: Thanks to the reviewers for their comments. I am so sorry that we forgot to focus on the integrity of the lines when drawing Figure 12. And now we have corrected the mistake, added to the manuscript and highlighted it.

    The location of the revision: Figure 14.

  1. 9. Comments:“The refractive index of sucrose solution ranges from 1.351 RIU to 1.4067 RIU.”However, Figure 14 shows the maximum RI of 1.402 instead of 1.4067;

Revision: We are very sorry for the mistake and have changed the index of refraction from 1.4067 to 1.402 in the manuscript.

    The location of the revision: The sixth line above figure 14. The sixth line above figure 17.

  1. 10. Comments:The format of the reference seems chaoticand needsto be modified;

Revision: Thank you for reminding me, and we have revised it according to the standard reference format, including adding the DOI, completing the author, bolding and italicizing the journal name.

    The location of the revision: The REFERENCE section.

  1. 11. Comments:I noticed some schemes in the Introduction showeven better sensitivity than the advantages of the sensor proposed by the authors. It is better to have a discussion or comparison to clear the merits and purpose of this manuscript.

Revision: Thank you for your suggestion. We also think it is necessary to compare the research results of the literatures. In order to make the manuscript more comprehensive, readable, and more in line with the requirements of Micromachines, the structures, response characteristics and disadvantages of various sensors have shown in Table 1. By comparison, it can be found that the sensor developed by us has the advantages of low price, high sensitivity, simple to make, easy to read and so on.

    The location of the revision: Table 1. The last sentence of the paragraph below Table 1.

   With best regards!

                                                            Yours faithfully

Reviewer 2 Report

A high sensitivity fiber optic Mach-Zehnder refractive index sensor based on waist-enlarged is proposed in this paper. The sensor improves spectral contrast and response sensitivity by fabricating a waist amplification double cone structure on the MZI arm of SMS. This paper explains the theoretical basis of the proposed new sensor, and gives the performance of the sensor and the corresponding experimental results. However, there are still some problems in the content and format of this paper. I think it can be published as an innovative paper after  overhaul. The specific review opinions are as follows:

  1. The name format of the picture in the article is not uniform, and some parts are left aligned and some are not left aligned.
  2. In the second part, the specific location parameters of the waist expansion on an optical fiber are not mentioned, and the conclusion part only describes the middle location, which is too vague to describe the expansion location.
  3. Inthe forth part , the pictures are not aligned.
  4. In Figs.14and 15, refractive index N blocks the spectral curve, which should be shown completely.
  5. In subsequent experiments, the new optical fiber was only compared with the optical fiber without waist enlargement structure, and the refractive index sensitivity performance was not compared with moving the waist enlargement part to other positions.

Author Response

Dear  reviewer:

Thank you very much for your comments about our paper submitted to High sensitivity optical fiber Mach-Zehnder refractive index sensor based on waist-enlarged bitaper (micromachines-1703654). After carefully studying the comments and your advice, we have made corresponding changes.

The following is the answers and revisions we have made in response to the reviewer's questions and suggestions item by item.

 A high sensitivity fiber optic Mach-Zehnder refractive index sensor based on waist-enlarged is proposed in this paper. The sensor improves spectral contrast and response sensitivity by fabricating a waist amplification double cone structure on the MZI arm of SMS. This paper explains the theoretical basis of the proposed new sensor, and gives the performance of the sensor and the corresponding experimental results. However, there are still some problems in the content and format of this paper. I think it can be published as an innovative paper after overhaul. The specific review opinions are as follows:

  1. Comments:The name format of the picture in the article is not uniform, and some parts are left aligned and some are not left aligned.

Revision: Thanks for your suggestion, we've settled the pictures in the middle and unified the format.

    The location of the revision: All the pictures in the manuscript.

  1. Comments:In the second part, the specific location parameters of the waist expansion on an optical fiber are not mentioned, and the conclusion part only describes the middle location, which is too vague to describe the expansion location.

Revision: Thank you for reading and suggestion. We truly think the questions you ask are very valuable. Because we have found that the position of the cone shows little impact on the refractive index, we didn’t discuss the phenomenon which makes the manuscript insufficient. Considering the problem you raised, we added the fiber waist-enlarged bitaper in one-half and one-third of the sensing arm to compare the difference. The specific position in the manuscript is as follows.

The location of the revision: The last sentence above figure 2. Figure 3, 7. Figure 15, 16. Figure 18, 19.

  1. Comments:In the forth part , the pictures are not aligned.

Revision: In the fourth part, we've settled the pictures in the middle and unified the format..

    The location of the revision: The pictures of the forth part .

  1. Comments:In Figs.14and 15, refractive index N blocks the spectral curve, which should be shown completely.

Revision: Thanks for your suggestion. We have modified the spectral charts in Figures 14 and 15, so that the spectrum can be presented clearly.

    The location of the revision: Figure 17 and 19.

    5. Comments:In subsequent experiments, the new optical fiber was only compared with the optical fiber without waist enlargement structure, and the refractive index sensitivity performance was not compared with moving the waist enlargement part to other positions.

Revision: Thanks for your suggestion. On the basis of the original manuscript, we added the fiber waist-enlarged bitaper in one-half and one-third of the sensing arm to compare the refractive index response. We found that no matter corrosion or not, the position of the waist-enlarged bitaper  has little effect on the sensor. So we add some pictures and sentences to describe this phenomenon.

The location of the revision: The last sentence above figure 14. Figure 14, 15, 16, 17, 18, 19.

   With best regards!

                                                           Yours faithfully

Round 2

Reviewer 1 Report

All the required revisions have been appropriately modified, the reviewer agrees to accept.